# Optimizing Cancer Treatment Through Gut Microbiome Modulation

**DOI:** 10.3390/cancers17071252

**Published:** 2025-04-07

**Authors:** Kyuri Kim, Mingyu Lee, Yoojin Shin, Yoonji Lee, Tae-Jung Kim

**Affiliations:** 1College of Medicine, Ewha Womans University, 25 Magokdong-ro 2-gil, Gangseo-gu, Seoul 03760, Republic of Korea; kkyuri01@ewhain.net; 2College of Medicine, The Catholic University of Korea, 222 Banpo-daero, Seocho-gu, Seoul 06591, Republic of Korea; mkl0107@catholic.ac.kr (M.L.); yoojins07@catholic.ac.kr (Y.S.); gina228@catholic.ac.kr (Y.L.); 3Department of Hospital Pathology, Yeouido St. Mary’s Hospital, College of Medicine, The Catholic University of Korea, 10, 63-ro, Yeongdeungpo-gu, Seoul 07345, Republic of Korea

**Keywords:** microbiome, cancer, immunotherapy, chemotherapy, probiotics, prebiotics, microbial metabolite

## Abstract

This review summarizes the impact of the gut microbiome on cancer treatment outcomes across three therapeutic approaches: immune checkpoint inhibitors, cytotoxic chemotherapy, and microbial interventions, including probiotics and fecal microbiota transplantation. We focus on five major cancer types—gastrointestinal cancer, lung cancer, liver cancer, breast cancer, and metastatic melanoma—to illustrate cancer type-specific microbiome associations. Particular emphasis is placed on microbial taxa and functional pathways that consistently influence treatment efficacy or toxicity across specific cancer types. By organizing preclinical and clinical evidence by therapy type and cancer type, this review offers a structured summary of current microbiome–cancer therapy interactions for researchers and clinicians.

## 1. Introduction

The microbiome has emerged as a critical determinant in cancer therapeutics, influencing responsiveness to both immunotherapy and chemotherapy. Intestinal microbial communities modulate immune responses and drug metabolism, directly impacting treatment efficacy and toxicity. While specific bacterial taxa enhance immune checkpoint inhibitor (ICI) activity, microbial dysbiosis can induce treatment resistance and adverse effects.

In chemotherapy, microbial enzymes activate or deactivate antineoplastic agents, altering therapeutic outcomes. Microbially derived metabolites, particularly short-chain fatty acids (SCFAs) and tryptophan-derived compounds, further regulate immune responses that affect treatment efficacy. Additionally, the microbiome plays a crucial role in drug metabolism, contributing to chemotherapy resistance and modulating drug toxicity.

Current research highlights microbiome modulation strategies in oncology, including fecal microbiota transplantation (FMT), probiotics, prebiotics, and postbiotics. These approaches, implemented alongside conventional treatments, show promise for enhancing therapeutic responses while reducing adverse effects.

As illustrated in Figure 1, the microbiome influences cancer therapy through three primary mechanisms: immune modulation, drug metabolism, and microbial metabolite production. It contributes to immune regulation by promoting T-cell activation, modulating ICI responses, and reducing immunosuppressive cells. Additionally, microbial communities influence chemotherapy efficacy by modulating drug activation, resistance mechanisms, and toxicity. Furthermore, microbial metabolites such as SCFAs and tryptophan-derived compounds play a key role in shaping the tumor microenvironment and immune response.

This manuscript analyzes the mechanisms underlying microbiome influence on cancer treatments across gastrointestinal, pulmonary, hepatocellular, breast, and metastatic melanoma malignancies. We evaluate current microbiome-based adjunctive therapies and their clinical applications, exploring the future development of microbiome-targeted interventions in oncological practice.

## 2. Enhancing Immunotherapy Through Microbiome Modulation

The gut microbiome exerts a critical influence on metabolic and immunological pathways that is essential for maintaining host equilibrium [1]. Beyond supporting immune homeostasis, it substantially impacts both localized and systemic antitumor immune responses [2]. This relationship is particularly pronounced in gastrointestinal (GI) cancers, where microbial dysbiosis—characterized by an imbalance in microbial populations—has been implicated in reduced therapeutic efficacy and adverse clinical outcomes [3].

Recent advances have highlighted distinct mechanisms by which the gut microbiota modulates immunotherapy. These include immune checkpoint regulation, tumor microenvironment remodeling, and the attenuation of therapy-associated adverse effects, as illustrated in Figure 2 [4,5,6]. Notably, several ongoing clinical trials are evaluating microbiome-based interventions to enhance immunotherapy efficacy, as summarized in Table 1. Specific microbial taxa, such as *Akkermansia muciniphila* and *Faecalibacterium prausnitzii*, have been identified as key contributors to enhancing the efficacy of immune checkpoint inhibitors (ICIs) [7,8,9]. Moreover, microbiota-derived metabolites, including short-chain fatty acids (SCFAs), play central roles in reducing inflammation, fostering T-cell infiltration, and reprogramming tumor-associated metabolic pathways [10]. Conversely, disruptions to the microbiome, often marked by the depletion of beneficial bacteria or overrepresentation of proinflammatory species, have been linked to suboptimal immunotherapy outcomes and increased susceptibility to immune-related adverse events (irAEs) [11].

Advances in microbiome research have opened new avenues for optimizing immunotherapy by both enhancing its therapeutic efficacy and mitigating associated toxicities. Figure 2 provides an overview of these mechanisms, underscoring the role of the microbiome in fine-tuning immune checkpoint pathways, reconfiguring the tumor microenvironment, and strengthening gut barrier function through the production of SCFAs. The following sections will delve deeper into these aspects, focusing on specific cancer contexts, including gastrointestinal, lung, liver, and breast cancers, as well as metastatic melanoma.

### 2.1. Gastrointestinal Cancer

Microorganisms inhabit various anatomical sites, with over 90% residing in the gastrointestinal (GI) tract [12]. The gut microbiota undergoes significant alterations in colorectal cancer (CRC), including reduced alpha diversity and disruption of microbial community structure [13]. Similar patterns of dysbiosis have been reported across GI malignancies. These shifts have been linked to variations in response to the immune checkpoint inhibitor (ICI), particularly anti–PD-1/PD-L1 agents [14].

Short-chain fatty acids (SCFAs), particularly butyrate, contribute to antitumor immunity by enhancing T cell cytotoxicity [15]. Enhanced anti-PD-1 efficacy has been observed in murine CRC models colonized with butyrate-producing taxa such as *Roseburia intestinalis*. This effect appears to be driven by increased intratumoral CD8^+^ T cell infiltration, resulting in reduced tumor burden [16]. Notably, in the MSS CT26 model, butyrate directly binds to TLR5 (K*_D_* = 264 μM), triggering NF-κB activation in cytotoxic T cells.

While these findings underscore the immunostimulatory potential of SCFAs, recent studies have revealed that their effects are not universally beneficial. In certain microbial and host contexts, SCFAs may instead promote tumor progression. For example, in CRC models colonized by *Fusobacterium nucleatum*, butyrate activates FFAR2, leading to Th17 cell expansion and IL-17-driven inflammation—an axis associated with tumor-promoting immune remodeling [17]. Similarly, in APC^*Min*/+^MSH2^−/−^ mice, butyrate enhances β-catenin signaling and drives epithelial transformation, highlighting that microbial metabolites may exert tumorigenic effects in genetically susceptible hosts [17]. This functional divergence is further reflected in population-level associations. In a case–control study, elevated plasma levels of acetic and propionic acid were positively associated with CRC occurrence (adjusted OR = 1.02 and 1.29; q = 0.007 and 0.03, respectively), whereas valeric and i-valeric acids showed inverse trends [18].

In addition to SCFAs, other microbial metabolites also contribute to immunomodulation. Inosine produced by *B. pseudolongum* enhances Th1 activation via the A2A receptor in murine CRC models, with its effect contingent on T cell costimulation, ultimately improving ICI efficacy [19].

Microbial structural components such as lipopolysaccharide (LPS) also exhibit strain-specific immune effects. LPS from *Fusobacterium periodonticum* (6 ng/mL) increased IL-1β, IL-6, and IFN-γ production in PBMCs, while LPS from *Bacteroides fragilis* and *Porphyromonas asaccharolytica* (600 ng/mL) suppressed cytokine levels (Cohen’s d > 0.8; *p* < 0.05) [20]. These divergent effects reflect differences in the immune contexture of CRC subtypes: CMS1 tumors are enriched with immunostimulatory bacteria, whereas CMS4 tumors are associated with immunosuppressive taxa.

Beyond CRC, similar microbiome-mediated modulation of ICI efficacy has been observed in other GI malignancies. In pancreatic cancer models, trimethylamine N-oxide (TMAO) improved anti-PD-1 response through enhanced immune activation [21]. In gastric cancer, a metagenomic study of patients with HER2-negative advanced disease receiving immunotherapy or combination treatment found that responders exhibited significantly higher relative abundance of *Lactobacillus*, particularly *L. salivarius* and *L. mucosae*, alongside increased microbial diversity. The abundance of these taxa correlated positively with PFS in both discovery and validation cohorts [22].

Taken together, microbial diversity, compositional features, and immunomodulatory metabolites collectively shape host responsiveness to ICI activity across gastrointestinal cancers.

### 2.2. Lung Cancer

In lung cancer, both gut and lung microbiota are increasingly recognized as modulators of immunotherapy outcomes [4,23]. The lung microbiota, in particular, is a key determinant of local immune responses, with dysbiosis being implicated in tumor progression and diminished therapeutic efficacy [23].

In advanced NSCLC patients receiving anti-PD-1 therapy (n=37), higher gut microbiota diversity has been associated with improved progression-free survival and favorable immunotherapy outcomes (median PFS: 209 vs. 52 days, *p* = 0.005) [24]. Responders were characterized by an enrichment of bacterial taxa such as *Alistipes putredinis*, *Bifidobacterium longum*, and *Prevotella copri*, which were present at baseline and further enriched during therapy [24].

Microbial imbalance may impair ICI responsiveness in lung cancer, mirroring observations in GI malignancies [11]. In contrast to GI tumors, where microbial modulation of the tumor immune microenvironment occurs locally via mucosal immune signaling [25], lung tumors are influenced primarily through systemic immune effects mediated by the gut–lung axis [23]. Ongoing clinical trials are exploring whether microbiome-modulating strategies, such as probiotic supplementation, can enhance ICI responsiveness in lung cancer patients (Table 1).

Several gut-derived taxa have been associated with favorable immunotherapy outcomes in NSCLC. These include members of the *Clostridia* class, *Bifidobacterium longum*, *Lactobacillus*, and *Phascolarctobacterium*, which have been linked to improved survival metrics such as progression-free survival and time to treatment failure [26,27,28]. However, it remains unclear whether these taxa preexisted or were enriched during therapy.

In line with these findings, SCFA-producing bacteria such as *Agathobaculum butyriciproducens* have also been linked to improved responses in KRAS-mutant NSCLC, possibly through enhanced inflammatory signaling [29].

Lung microbiota composition, including the enrichment of oral commensals such as *Veillonella*, *Prevotella*, and *Streptococcus*, has been associated with chronic inflammation and immune evasion [23,30]. These taxa, several of which are known oral commensals, may impair cytotoxic cell function and activate protumor signaling pathways [31]. Tobacco exposure, a major environmental modifier of the lung microbiota, promotes dysbiosis characterized by the expansion of taxa such as *Odoribacter*, *Alistipes*, and *Ruminococcus*, as well as the depletion of SCFA-producing species like *Akkermansia* [32]. Evidence from COPD cohorts—though derived from non-malignant populations—suggests that similar microbial shifts, including increased *Streptococcus* and *Haemophilus*, may recapitulate immune-modulatory patterns relevant to lung cancer [33].

Altogether, findings from both gut- and lung-focused studies point to a multifaceted role of the microbiota in shaping immunotherapeutic outcomes in lung cancer. Through systemic signaling pathways originating in the gut and local immune modulation within the lung, microbial communities help shape the tumor microenvironment and are increasingly recognized as promising targets for therapeutic modulation.

### 2.3. Liver Cancer

The interactions between microbial composition, bile acid metabolism, and the immune microenvironment are critical in hepatocellular carcinoma (HCC) pathogenesis, particularly in non-alcoholic fatty liver disease (NAFLD) [34]. The liver, connected to the gut via portal circulation, is influenced by alterations in the gut microbiota that impact immune responses and disease progression. These microbial changes modulate ICI efficacy, potentially improving treatment outcomes.

NAFLD-associated HCC is distinguished from its viral counterpart by its distinct metabolic and immunological characteristics. Chronic inflammation driven by obesity and diabetes fosters an immunosuppressive environment, which may undermine the efficacy of ICI therapy [35]. Moreover, NAFLD-associated HCC exhibits unique immune evasion mechanisms that differ from viral HCC, necessitating tailored therapeutic approaches. Although ICI therapy has demonstrated therapeutic potential, NAFLD-associated immunosuppression remains a significant barrier, necessitating further investigation into its underlying mechanisms.

Dysbiosis, frequently observed in NAFLD, increases intestinal permeability, allowing microbial products such as LPSs to translocate into the portal circulation. This promotes chronic hepatic inflammation and fibrosis, which are key contributors to HCC progression [35,36]. Additionally, alterations in bile acid metabolism influence immune homeostasis, further contributing to an immunosuppressive tumor microenvironment.

Recent therapeutic strategies aim to restore microbial balance and modulate bile acid metabolism. FXR agonists, engineered probiotics, and dietary modifications are under investigation for their potential to enhance ICI response [36]. Several clinical trials are evaluating microbiome-based interventions in HCC. For example, NCT06551272 is investigating the use of *Faecalibacterium prausnitzii*-derived EXL01 in combination with atezolizumab and bevacizumab (Table 1). Similarly, NCT05620004 is examining whether *Bifidobacterium bifidum* can enhance the effects of carrilizumab and apatinib mesylate in advanced HCC patients (Table 1).

Despite these promising developments, substantial challenges persist. The heterogeneity of NAFLD-associated HCC, coupled with the complex interplay between gut microbiota and bile acid metabolism, requires comprehensive and longitudinal studies to identify actionable targets. Tailored therapeutic strategies that address metabolic comorbidities and microbiota diversity are critical for advancing clinical outcomes in this patient population.

### 2.4. Breast Cancer

Breast cancer is generally classified as an immune-cold tumor characterized by low immune cell infiltration and an immunosuppressive microenvironment, which limits responsiveness to ICIs [37]. Unlike immunogenic tumors that attract effector T cells, breast cancer exhibits poor immune infiltration driven by active immunosuppressive mechanisms in the TME [37]. These mechanisms include TGF-β signaling, tumor-associated macrophages (TAMs), regulatory T cells (Tregs), and myeloid-derived suppressor cells (MDSCs), which collectively inhibit antitumor immune responses and reduce the efficacy of ICIs.

The gut microbiota profoundly affects systemic immune responses and tumor progression. Enzymatic activity, such as β-glucuronidase (GUS), facilitates the breakdown of estrogen metabolites, increasing their systemic availability [38]. Elevated estrogen levels are associated with tumor proliferation and the establishment of an immunosuppressive TME through the activation of Tregs and MDSCs, further diminishing the effectiveness of ICIs [38].

Targeting GUS activity demonstrates promise as a therapeutic approach. Inhibitors such as UNC10201652 have been shown to reduce estrogen reactivation, thereby counteracting immunosuppressive mechanisms within the tumor milieu and augmenting ICI responses [38]. Although primarily studied in hormone receptor-positive breast cancer, the prospective role of these strategies in other subtypes remains under investigation.

The breast tissue microbiota is another factor that facilitates tumor progression and immune modulation. Distinct microbial communities have been identified in both healthy and cancerous breast tissue, with tumor samples frequently enriched in bacterial genera such as *Ralstonia*, *Methylobacterium*, and *Sphingomonas*, which are associated with metabolic alterations that support tumor survival [39]. While their specific contribution to immune modulation has yet to be fully elucidated, these bacteria may influence immune cell recruitment and function within the TME.

The composition of the gut microbiota is a critical determinant of ICI efficacy in breast cancer. Reduced microbial diversity, including lower levels of Lachnospiraceae and Bifidobacteriaceae, is associated with resistance to trastuzumab in HER2-positive patients [40]. Conversely, beneficial bacteria such as *Akkermansia muciniphila* augment ICI responses by restoring CD8^+^ T cell cytotoxicity and promoting immune surveillance, particularly in tumors with low PD-L1 expression [7]. Additionally, metabolites such as SCFAs, produced by *Faecalibacterium* and *Bacteroides*, optimize immune function and enhance treatment efficacy [41].

Dietary modifications underscore the potential of microbiota-targeted interventions in breast cancer. In preclinical models of TNBC, quercetin supplementation has been shown to increase the abundance of *Akkermansia muciniphila*, correlating with improved responses to anti-PD-1 therapy and cyclophosphamide. These effects have been attributed to heightened NK cell activity and a reduction in Tregs [8].

Although the precise function of bacterial genera such as *Ralstonia* and *Methylobacterium* has yet to be fully elucidated, their enrichment in tumor tissues suggests that they could serve as potential biomarkers for disease progression or therapeutic targets for modulating the TME. Further investigation is required to determine their precise impact on immune dynamics and whether their presence correlates with treatment outcomes in breast cancer patients.

### 2.5. Metastatic Melanoma

Metastatic melanoma is an aggressive malignancy associated with high mortality rates and historically limited therapeutic options. The introduction of immunotherapy, particularly ICIs, has led to substantial improvements in survival outcomes for patients with advanced disease [42].

The gut microbiome is instrumental in modulating the efficacy of ICIs in metastatic melanoma. Specific bacterial taxa, such as *Bacteroides fragilis*, have been shown to enhance the therapeutic effects of ipilimumab, a CTLA-4 inhibitor, by influencing host immune responses [43]. Likewise, the oral administration of *Bifidobacterium* species in murine melanoma models has been demonstrated to improve tumor control in conjunction with anti-PD-L1 therapy [44].

Gut microbiota composition influences ICI responses: the enrichment of *Faecalibacterium* and Firmicutes is associated with prolonged PFS and OS—in contrast to *Bacteroides* profiles [45].

Increased microbial diversity and the presence of SCFA-producing bacteria, such as *Faecalibacterium prausnitzii*, *Ruminococcaceae*, and *Akkermansia muciniphila*, have been associated with favorable immunotherapy outcomes across melanoma cohorts [9,46]. These taxa enhance antitumor immunity through SCFA-mediated T cell activation and barrier integrity. However, the role of SCFAs is context-dependent. Certain species, such as *Bacteroides fragilis*, have been implicated in immune-related adverse events (irAEs) via IL-1β-mediated inflammation, while others may exhibit tumor-promoting effects under specific host or microbial conditions [47]. This functional duality underscores the need to evaluate microbial metabolites within the broader ecological and immunological context of each patient.

Microbiota-based models incorporating bacterial and eukaryotic taxa have demonstrated strong performance in predicting ICI responsiveness, with an AUROC of 0.8019 in metastatic melanoma [9]. Yet, meta-analyses suggest that interindividual microbiome variability limits the reliability of universal biomarkers, as reflected by a lower AUROC of 0.625 [48].

Intratumor microbiome diversity in TCGA melanoma samples has also been correlated with ICI responsiveness, with taxa such as *Eudoraea* enhancing CD8^+^ T cell infiltration [49]. Intratumor microbial features may reflect local immune context and hold promise as predictive biomarkers.

Longitudinal analyses have revealed distinct microbiome shifts during ICI therapy, with responders showing the enrichment of beneficial taxa such as *Faecalibacterium prausnitzii*, *Akkermansia muciniphila*, *Agathobaculum butyriciproducens*, and *Lactobacillus gasseri*. Notably, taxa such as *A. butyriciproducens* persisted following therapy, while *S. intestinalis* and *B. clarus*, enriched in non-responders declined to baseline levels [50].

Among the microbial candidates, *A. muciniphila* and *F. prausnitzii* stand out for their consistent association with clinical response and mechanistic validation [51]. Computational models incorporating these taxa achieved high predictive accuracy [9]. Ongoing trials are evaluating *A. muciniphila* (Oncobax-AK) and *F. prausnitzii* (EXL01) as adjunctive therapies alongside ICIs (Table 1; NCT05865730 and NCT06448572). Together with gut-derived signals, intratumor microbial signatures expand the spectrum of clinically translatable biomarkers for melanoma immunotherapy.

In conclusion, both gut and intratumor microbiota constitute clinically actionable biomarkers and modulators of ICI efficacy in metastatic melanoma. SCFA-producing taxa enhance antitumor immunity, whereas proinflammatory microbes may exacerbate toxicity. Future efforts should refine microbiota-based predictors and validate their clinical utility in guiding ICI selection and individualizing immunotherapy.

## 3. Enhancing Chemotherapy Through Microbiome Modulation

Chemotherapy remains a cornerstone of cancer treatment, yet its efficacy is often compromised by systemic toxicity, therapeutic resistance, and significant variability in patient responses. Recent studies demonstrate that gut microbiota significantly contribute to these outcomes by regulating drug metabolism, inflammatory pathways, and tumor microenvironment dynamics [52,53]. Accordingly, a number of clinical trials have been initiated to evaluate microbiome-modulating strategies alongside chemotherapy across various cancer types (Table 2).

As shown in Figure 3, the microbiome’s influence on chemotherapy can be categorized into three key domains: drug metabolism, inflammation and toxicity, and chemotherapy resistance. For instance, microbial enzymes can degrade chemotherapeutic agents or reactivate toxic metabolites, altering their pharmacological efficacy [54,55]. Similarly, dysbiosis exacerbates systemic inflammation through LPS-driven pathways, heightening treatment-associated toxicity. Moreover, microbial interactions within the tumor microenvironment contribute to resistance mechanisms by promoting autophagy, inhibiting apoptosis, or impeding drug penetration [56].

This framework assesses how the microbiome modulates chemotherapy response through distinct biological mechanisms. The following sections will investigate these relationships across different cancer types, with a particular focus on microbiota-driven influences on treatment efficacy and their implications for optimizing therapeutic strategies.

### 3.1. Gastrointestinal Cancer

Chemotherapy remains a cornerstone in the management of GI malignancies; however, its therapeutic efficacy is frequently limited by resistance and toxicity [57].

Accumulating evidence demonstrates that gut microbes can interfere with treatment efficacy through the enzymatic biotransformation of drugs and modulation of host immunity [55,58]. For example, microbial cytidine deaminase converts gemcitabine into its inactive metabolite, 2’,2’-difluorodeoxyuridine. In murine models of CRC colonized with *Gammaproteobacteria*, gemcitabine monotherapy failed to suppress tumor growth, whereas cotreatment with ciprofloxacin led to a nearly 60% reduction in tumor volume [58]. Similarly, microbial β-glucuronidase reverses the hepatic glucuronidation of irinotecan, reactivating SN-38 and promoting severe gastrointestinal toxicity, including diarrhea [55].

Beyond enzymatic activity, microbial composition shapes the tumor microenvironment (TME). In CRC patients, high intratumoral *Fusobacterium nucleatum* was significantly associated with disease recurrence and inferior recurrence-free survival (AUC = 0.875) [59]. In vivo, *F. nucleatum* attenuated the tumor-suppressive effects of 5-fluorouracil (5-FU)—an effect that was reversed by BCL2 knockdown [59]. This bacterium contributes to chemoresistance through multiple mechanisms, including the inhibition of chemotherapy-induced pyroptosis via Hippo-YAP-mediated BCL2 upregulation and the enhancement of autophagy through TLR4 signaling and miRNA modulation [59,60].

These findings exemplify the role of “complicit microbes”—a group of pathobionts that promote tumor progression through chronic inflammation, immune evasion, and metabolic reprogramming [61]. Notably, colibactin-producing *E. coli* (CoPEC) reshapes the TME by inducing lipid droplet accumulation, thereby limiting CD8^+^ T cell infiltration and IFN-γ production [62].

While chemotherapy exerts substantial perturbations on the gut microbial ecosystem, certain beneficial taxa appear to re-emerge in patients exhibiting favorable therapeutic responses. In a prospective study of stage IV CRC patients receiving XELOX chemotherapy, the relative abundance of *Bifidobacterium longum* significantly increased post-treatment (*p* < 0.05) [63]. Notably, this increase was more pronounced in individuals with stable disease compared to those with progressive disease (*p* = 0.023), suggesting a potential role for microbial reconstitution in sustaining treatment benefit.

Chemotherapy-driven microbiota shifts may either exacerbate toxicity or support therapeutic benefit, depending on the direction of microbial reassembly. Understanding these dynamics may inform microbiota-targeted strategies to improve therapeutic efficacy and mitigate toxicity.

### 3.2. Lung Cancer

NSCLC is frequently diagnosed at advanced or metastatic stages, often precluding surgical interventions. For these patients, platinum-based doublet chemotherapy, typically combining cisplatin with gemcitabine or paclitaxel, remains the standard therapeutic approach [64].

The gut–lung axis, a two-way communication network connecting the GI and respiratory systems, regulates systemic immune responses and localized inflammation [65]. Microbial dysbiosis is linked to weakened immune responses and reduced chemotherapy efficacy [65]. Furthermore, antibiotic-induced microbiota depletion has been associated with OS and PFS in NSCLC patients undergoing chemotherapy [66].

SCFAs, as mentioned earlier, influence chemotherapy responses by modulating the immune system and inflammation. In a murine NSCLC model, concurrent antibiotic use negated the tumor-reducing effects of cisplatin. Conversely, oral supplementation with *Lactobacillus acidophilus* enhanced cisplatin’s efficacy, resulting in greater tumor reduction and improved survival outcomes [67]. This effect was linked to changes in genes regulating angiogenesis (*VEGFA*), apoptosis (*BAX*), and cell cycle progression (*CDKN1B*) [67].

While baseline microbiome composition correlates with chemotherapy response, certain bacterial species may actively enhance treatment efficacy through direct metabolic and immune modulation. A metagenomic study has identified distinct bacterial taxa linked to chemotherapy response in NSCLC patients, providing further insight into host–microbiome interactions [68]. In a cohort of advanced-stage cases, microbial profiling revealed substantial differences between individuals with favorable treatment outcomes and those with limited therapeutic benefits. Responders exhibited a greater prevalence of *Streptococcus mutans* and *Enterococcus casseliflavus*, whereas individuals with diminished efficacy exhibited higher levels of *Leuconostoc lactis* and *Eubacterium siraeum* [68]. Further metabolic assessments indicated that responders exhibited increased L-glutamate degradation, while non-responders relied more heavily on carbohydrate fermentation [68].

Chemotherapy itself induces substantial shifts in gut microbial composition over the course of treatment. Longitudinal monitoring of gut microbiota before and after chemotherapy revealed marked compositional alterations, including an increased prevalence of *Firmicutes*, *Euryarchaeota*, and *Synergistetes*, alongside a reduction in *Bacteroides*, *Proteobacteria*, and *Actinobacteria* [69]. Notably, chemotherapy-induced gastrointestinal toxicity was linked to an increased abundance of *Prevotella*, *Megamonas*, *Streptococcus*, and *Faecalibacterium*, whereas patients with higher levels of *Veillonella*, *Ruminococcus*, and *Akkermansia* exhibited lower toxicity rates. These observations underscore the intricate relationship between pre-existing microbiome composition and chemotherapy-induced alterations, influencing both therapeutic response and treatment-related complications [69].

Advancing microbiome-based therapeutic strategies requires a deeper understanding of the microbiota-driven metabolic pathways that influence drug bioavailability and immune modulation. Defining specific microbial taxa that predict chemotherapy response and incorporating microbiome-based biomarkers into clinical workflows may enhance precision oncology approaches.

### 3.3. Liver Cancer

Hepatocellular carcinoma presents significant challenges for chemotherapy due to its intrinsic drug resistance and the critical role of hepatic metabolism in drug clearance. Unlike lung cancer, where systemic chemotherapy remains a primary strategy, HCC treatment often integrates locoregional therapies such as transarterial chemoembolization (TACE) and hepatic artery infusion chemotherapy (HAIC) to improve drug delivery and efficacy. However, chemotherapy resistance remains a major limitation, necessitating novel approaches to enhance treatment outcomes [70].

The gut microbiota significantly influences HCC chemotherapy by modulating bile acid metabolism, drug detoxification, and systemic inflammation. Dysbiosis alters the gut–liver axis, disrupting bile acid homeostasis and leading to an immunosuppressive tumor microenvironment. This disruption has been attributed to the reduction in bile salt hydrolase (BSH)-producing taxa, including *Bifidobacteriales* and *Clostridiales*, which impairs the conversion of primary to conjugated secondary bile acids, such as glycodeoxycholic acid (GDCA). Importantly, GDCA exerts antiproliferative and proapoptotic effects on HCC cells, indicating that its depletion contributes to a tumor-promoting microenvironment and may reduce chemosensitivity [71]. Such dysregulation of the bile acid pool may ultimately impair the metabolism and efficacy of chemotherapeutic agents, particularly fluoropyrimidines and anthracyclines, which rely on hepatic enzymatic activation [72].

Natural products (NPs), including flavonoids and triterpenoids, have been studied for their role in enhancing chemotherapy responses in HCC. Flavonoids suppress tumor proliferation by inhibiting the Wnt/β-catenin signaling pathway, whereas triterpenoids, such as ginsenosides, attenuate NF-κB-mediated inflammation, which is a key driver of chemotherapy resistance [73]. Recent studies suggest that certain triterpenoids, such as those derived from *Ganoderma lucidum*, may influence gut microbiota composition, contributing to the suppression of liver cancer progression [73]. While the direct impact of flavonoids and triterpenoids on gut microbiota remains under investigation, their ability to regulate inflammatory pathways and bile acid metabolism may indirectly support a more favorable gut–liver axis for chemotherapy efficacy.

Building on these insights, microbiota-targeted interventions represent a promising avenue for overcoming chemotherapy resistance in HCC. Such strategies include dietary modulation, microbial metabolite supplementation, and engineered bacterial therapies that aim to restore microbial functions, including BSH activity [73]. Notably, the targeted restoration of GDCA levels has demonstrated therapeutic efficacy in preclinical models, where oral GDCA administration suppressed tumor growth and promoted apoptosis in HCC-bearing mice [71].

### 3.4. Breast Cancer

Chemotherapy is a cornerstone treatment for breast cancer, particularly for triple-negative and high-risk hormone receptor-positive subtypes. Neoadjuvant chemotherapy, administered prior to surgery to shrink tumors, offers valuable insights into treatment sensitivity. However, its efficacy varies among molecular subtypes, with higher pathological complete response rates observed in triple-negative and HER2-positive cancers compared to hormone receptor-positive tumors [74].

In patients with triple-negative breast cancer, those achieving a pathological complete response after neoadjuvant chemotherapy showed significantly higher gut microbial α diversity [75]. Further analysis identified specific bacterial taxa, notably *Bacteroides eggerthii*, which were more abundant in treatment responders [75].

Certain gut bacterial enzymes, particularly β-glucuronidases, influence tamoxifen metabolism by reactivating its inactive metabolites, thereby enhancing its bioavailability and therapeutic efficacy. Consequently, excessive inhibition of β-glucuronidases could reduce metabolite reactivation, compromising tamoxifen’s clinical effectiveness [76,77]. Additionally, bacterial enzymatic activity can directly degrade chemotherapeutic agents, reducing their antineoplastic effects. For example, *Gammaproteobacteria*-derived cytidine deaminase inactivates gemcitabine, compromising its efficacy. This effect has been reversed in preclinical models using ciprofloxacin to inhibit bacterial cytidine deaminase, restoring gemcitabine’s anticancer activity [78].

Changes in microbial community composition not only influence chemotherapy response but also play a critical role in resistance development and toxicity profiles. For example, increased abundance of *Veillonella* correlates with resistance to aromatase inhibitors, potentially through the modulation of estrogen-deconjugating enzymatic activities that alters estrogen metabolism pathways [79]. Future studies should aim to identify precise microbial biomarkers that are predictive of therapy responses and elucidate their mechanistic roles in drug metabolism and resistance.

The gut microbiome also significantly contributes to chemotherapy-induced toxicity, primarily through the microbial modulation of drug metabolism. For instance, irinotecan’s active metabolite, SN-38, is reactivated by bacterial β-glucuronidases, resulting in severe GI toxicities such as mucosal damage and diarrhea. Preclinical studies demonstrated that inhibiting these microbial enzymes effectively reduces SN-38 reactivation and mitigates irinotecan-induced toxicity [6,78].

In conclusion, the gut microbiome substantially impacts chemotherapy efficacy, resistance, and toxicity in breast cancer. A comprehensive understanding of microbiome–drug interactions and their underlying mechanisms could facilitate the development of targeted microbiome-based strategies, ultimately enhancing treatment outcomes and advancing personalized oncology.

### 3.5. Metastatic Melanoma

Advanced metastatic melanoma presents formidable therapeutic challenges, particularly when resistance to ICIs or targeted therapies manifests. Although chemotherapeutic approaches occupy a subordinate position relative to immunotherapeutic interventions, they maintain clinical relevance in select circumstances. Accumulating evidence indicates that the *gut microbiota* exerts significant influence on chemotherapeutic efficacy and associated toxicity through the modulation of immune function and transformation of the tumor microenvironment [80].

Specific chemotherapeutic compounds, notably cyclophosphamide (CTX), induce substantial alterations in *gut microbiota* composition characterized by diminished populations of beneficial bacterial taxa, including *Faecalibacterium prausnitzii* and *Roseburia* [52]. These microbial disruptions compromise intestinal barrier function, thereby facilitating bacterial translocation and consequent immunological activation [52]. This process operates through MyD88-dependent signaling mechanisms, resulting in enhanced Th1 and Th17 immunological responses that potentially influence therapeutic outcomes [52]. Furthermore, CTX administration modifies dendritic cell functionality by augmenting antigen acquisition and T-lymphocyte priming via the TLR/MyD88/MAPK signaling axis, further potentiating Th1 and Th17 cellular differentiation [81].

The microbial compositional shifts associated with chemotherapeutic intervention extend beyond localized effects to impact the broader tumor microenvironment [80]. Diminished SCFA concentrations contribute to immunological dysregulation, potentially influencing therapeutic response, resistance mechanisms, and treatment-related toxicity [80]. Additionally, alterations in microbial community structure demonstrate associations with epithelial-to-mesenchymal transition (EMT) and angiogenic processes, which are both recognized as significant factors in tumor progression dynamics [80].

While CTX-induced *gut microbiota* alterations have been extensively characterized, the interactions between alternative chemotherapeutic agents employed in melanoma management, particularly dacarbazine, and the intestinal microbiome remain inadequately investigated [80]. Contemporary research suggests that microbiota composition may significantly influence chemotherapy-induced immunological responses, though additional investigation is warranted to elucidate the precise mechanisms by which these microbial alterations impact therapeutic efficacy.

Investigative efforts continue to examine potential microbiome modulation strategies to enhance chemotherapeutic responses in melanoma treatment contexts [80]. Comprehensive understanding of these intricate host–microbiome interactions may reveal novel therapeutic opportunities through the integration of microbiota-targeted interventions with established treatment protocols, potentially improving clinical outcomes in this challenging malignancy.

## 4. Microbiome-Driven Supportive Interventions in Cancer Treatment

Dysbiosis, referring to the imbalance of microbial communities, is frequently observed in cancer patients and has been implicated in both tumor progression and therapeutic response [13,23]. However, whether dysbiosis acts as a cause or consequence of cancer remains debated. Some studies suggest that it contributes to oncogenesis by promoting inflammation, genomic instability, and immune evasion [82,83], whereas others view it as a secondary effect driven by tumor-induced metabolic changes, immune modulation, or therapy-associated perturbations [84].

Given the association between dysbiosis and treatment outcomes, various microbiome-based strategies have been explored to restore microbial homeostasis, including fecal microbiota transplantation (FMT), antibiotics, and microbiota-directed biotherapeutics. Among the latter, *prebiotics*, *probiotics*, and *postbiotics* differ in both their biological mechanisms and timing of administration. *Prebiotics* are generally administered *before or during* cancer therapy to promote the growth of beneficial microbes and preserve microbial diversity. *Probiotics*, consisting of live microorganisms, are typically used *concurrently* with chemotherapy or immunotherapy to reduce gastrointestinal toxicity, reinforce the intestinal barrier, and modulate systemic immune responses. In contrast, *postbiotics*—non-viable microbial products or metabolites—are introduced *during or after* treatment, offering anti-inflammatory and immunomodulatory effects while minimizing the safety concerns associated with live microbial administration in immunocompromised patients [85].

The following sections examine the current landscape of these interventions and their potential to enhance therapeutic efficacy in oncology.

### 4.1. FMT

Fecal microbiota transplantation (FMT), a microbiome-based intervention, introduces beneficial microbial communities from healthy donors [86] to restore gut homeostasis and modulate host responses during cancer treatment [87,88].

Ongoing clinical trials continue to explore the potential of FMT in modulating immunotherapy response across different cancer types. As shown in Table 3, FMT is being investigated in multiple malignancies. Trials employ various administration routes, such as oral capsules and colonoscopic infusion. A key focus has been on transferring microbiota from ICI responders to non-responders to overcome therapeutic resistance through microbial modulation. For instance, FMT from immunotherapy responders may enhance efficacy in previously resistant metastatic melanoma patients [89], which is supported by preclinical findings confirming restored responsiveness to PD-1 blockade and immune infiltration modulation [5].

Additionally, FMT has been investigated as a strategy to mitigate chemotherapy-induced dysbiosis. Such microbial interventions have been shown to persist for several months post-transplantation, supporting microbiome restoration and improved chemotherapy tolerance [90,91]. In one study, donor-derived microbial strains were detected in recipients up to three months after FMT, with strain-level coexistence confirmed by single-nucleotide variant tracking [91].

Most current FMT trials in oncology remain in early-phase development, in part because clinical implementation is constrained by safety concerns and inconsistencies arising from donor-to-donor microbial variability, which together underscore the urgent need for standardization.

Although generally well-tolerated, FMT carries risks, particularly infections linked to upper gastrointestinal administration, posing greater concerns for immunocompromised oncology patients [92,93]. Thus, the route selection and patient condition must be carefully considered in clinical practice. In particular, donor-to-donor variation in microbiota composition remains a major barrier to reproducible clinical outcomes. The therapeutic success of FMT is highly dependent on donor-specific microbial composition, particularly strain-level differences that govern engraftment dynamics and long-term microbiome stability in the recipient [94]. Recent strategies to address this include synthetic microbial consortia and microbial fingerprinting. While synthetic microbial consortia reduce variability by introducing a standardized set of defined strains, microbial fingerprinting focuses on optimizing donor selection through the identification of favorable microbial signatures [95]. Future efforts should prioritize translating these strategies into clinically applicable protocols—backed by long-term safety monitoring, regulatory approval pathways, and robust efficacy data across diverse patient populations.

### 4.2. Prebiotics

Prebiotics are defined as “substrates selectively utilized by host microorganisms to confer health benefits” [96]. Rather than introducing external microbial strains, these compounds promote beneficial resident bacteria that support mucosal immunity and epithelial integrity [96,97].

Gut microbiota modulation by prebiotics has been explored as a complementary approach to existing cancer treatments. In one study, inulin-based hydrogels loaded with oxaliplatin and MnO_2_ nanoparticles (Oxa@HMI) improved metabolite profiles, restored microbial balance, and enhanced antitumor immune responses in preclinical models [10]. Specifically, inulin alone was found to expand the abundance of *Akkermansia muciniphila*, which is a species linked to epithelial barrier reinforcement and decreased systemic inflammation [97]. In addition, plant-derived compounds with prebiotic activity have been shown to increase SCFA-producing genera such as *Blautia*, which are inversely correlated with inflammation and metabolic dysregulation [98]. Nutritional interventions, such as black raspberry supplementation, have shown potential to reshape microbial composition and mitigate tumorigenic factors [99].

Prebiotics have been incorporated into chemotherapeutic formulations to enhance drug bioavailability and microbiota-mediated efficacy. For example, a xylan-based capecitabine complex (SCXN) enhanced drug bioavailability and simultaneously supported the proliferation of bacterial taxa such as *Akkermansia* and *Faecalibaculum*, which are associated with improved treatment efficacy in murine models [100].

While these preclinical results are encouraging, the clinical translation of prebiotics remains challenging. Interindividual variability in microbiome composition significantly alters responses to prebiotic interventions, limiting reproducibility and standardization across populations [101,102]. In addition, the need for high therapeutic doses and the susceptibility of outcomes to external confounders such as diet and medication use further complicate trial design and scalability [102].

These limitations highlight the need for personalized prebiotic strategies tailored to host microbial and dietary profiles. Therapeutic outcomes may be improved through selective enrichment of taxa such as *Akkermansia* or SCFA-producing genera guided by predictive microbial biomarkers and integrated into precision-based treatment frameworks [103].

### 4.3. Probiotics

Probiotics constitute “live strains of microorganisms that, when administered in adequate amounts, confer health benefits to the host” [96]. Within oncological contexts, these microbial interventions facilitate the restoration of microbial homeostasis, suppress complicit microbes which directly contribute to tumorigenesis [61], and generate critical bioactive compounds, notably SCFAs, which play instrumental roles in inflammatory modulation and intestinal barrier maintenance [10].

Probiotics such as *Bifidobacterium* species have been proposed to reduce carcinogenic metabolite toxicity and enhance antitumor immune responses [44,64]. Preclinical investigations reveal that oral *Bifidobacterium* supplementation may potentially enhance anti-PD-L1 efficacy through immune priming and increased tumor-infiltrating T cell responses [44]. Clinical evidence from meta-analyses among NSCLC patients receiving ICIs indicates that probiotic supplementation was associated with improved overall survival (HR = 0.50, 95% CI: 0.30–0.85) and progression-free survival (HR = 0.51, 95% CI: 0.42–0.61) [104].

Furthermore, genetically engineered probiotic organisms present innovative strategies for TME modulation. For instance, oral administration of IL-2-expressing attenuated *Salmonella* strains led to increased NK and NK-T cell activity in a phase I trial [105]. Unlike prebiotics, which rely on host-dependent microbial fermentation and exhibit considerable interindividual variability [101,102], engineered probiotics can be designed to deliver therapeutic payloads directly within the tumor microenvironment, including cytokines such as IL-2 [105], thereby enabling more targeted and controllable immunomodulation.

Despite increasing clinical trials and growing evidence supporting the potential of probiotic-based interventions, several barriers hinder their clinical translation. First, safety concerns persist, as rare but serious cases of bacteremia and sepsis have been reported following the administration of probiotics such as *Lactobacillus rhamnosus GG* and *Saccharomyces boulardii* [106,107]. Second, the potential for horizontal transfer of antibiotic resistance genes (ARGs) between probiotic strains and pathogenic bacteria raises significant biosafety issues [106]. Third, variability in strain viability and host colonization efficiency poses challenges to consistent therapeutic delivery [108]. Finally, regulatory frameworks for live biotherapeutic products remain underdeveloped, with strain-specific characterization and manufacturing controls adding further complexity to clinical implementation [102].

To bridge these translational gaps, future efforts should prioritize the development of standardized evaluation frameworks that balance therapeutic efficacy with biosafety. Strain-specific selection based on immunological function, resistance profiles, and colonization capacity must be guided by rigorous preclinical and clinical validation. Ultimately, the safe and effective integration of probiotics into oncology care will depend on harmonized regulatory pathways and precision-based deployment strategies tailored to individual patient and microbial characteristics.

### 4.4. Postbiotics

Postbiotics, metabolic byproducts of microbial activity have gained attention for their immunomodulatory properties and potential application as adjuncts in cancer therapy. Unlike probiotics, which require bacterial colonization, postbiotics exert their effects through bioactive compounds such as SCFAs, tryptophan-derived metabolites, exopolysaccharides (EPSs), and microbial cell wall fragments [109,110]. Such characteristics render postbiotics promising candidates for modulating the tumor microenvironment and enhancing immunotherapy efficacy.

SCFAs, including butyrate, propionate, and acetate, are known to regulate immune responses and influence ICI therapy outcomes [14]. These metabolites impact both Treg and CD8^+^ T cells, contributing to enhanced antitumor activity through histone deacetylase (HDAC) inhibition [111,112,113]. Notably, butyrate increases IL-12R expression in CD8^+^ T cells by upregulating inhibitor of DNA Binding 2 (ID2), further strengthening antitumor immunity [103]. However, the rapid clearance of SCFAs from circulation remains a challenge, leading to the development of nanoparticle-based delivery strategies to improve their bioavailability [114,115].

SCFAs do not exert uniform effects across different ICI therapies. Higher fecal propionate levels have been associated with better responses to anti-PD-1 therapy, whereas increased systemic butyrate concentrations have been linked to reduced efficacy of anti-CTLA-4 therapy [116,117]. Additionally, butyrate may promote immunosuppressive effects, such as expanding Tregs and increasing IL-10 production, under certain host-dependent contexts [118]. These divergent outcomes emphasize that SCFA-mediated immune modulation is influenced by factors such as treatment timing, metabolite concentration, and the local immune microenvironment [117,118].

Beyond SCFAs, tryptophan-derived metabolites also play a key role in immune regulation. Indole-3-aldehyde, produced by *Lactobacillus reuteri*, has been shown to enhance ICI efficacy by stimulating IFN-γ production in CD8^+^ T cells [15].

Exopolysaccharides (EPSs) and other microbial-derived components further contribute to immune modulation. EPSs from *Bacillus coagulans* have been found to reshape the tumor-associated microbiome, improving gut homeostasis and reducing colorectal cancer progression [119]. Additionally, metabolites from *Lactobacillus* strains influence oncogenic pathways by modulating cell proliferation, apoptosis, and inflammation [120].

Despite their therapeutic potential, the variability of individual gut microbiota composition remains a challenge for postbiotics, similar to other microbial interventions such as probiotics and FMT [121]. However, recent evidence suggests that postbiotics might also exert colonization-independent effects—a distinct advantage over live microbial interventions such as probiotics and FMT. For instance, metabolites derived from *Bifidobacterium breve* and *Lactobacillus rhamnosus* induced apoptosis in colorectal cancer cells by modulating the expression of apoptosis-related genes (increased Bax and caspase-3 and decreased Bcl-2) and suppressed metastasis-associated genes (RSPO2, NGF, and MMP7) [122]. Acting via defined molecular routes, postbiotics may complement cancer therapy while providing a targeted adjunctive option that is less dependent on host microbiota composition.

### 4.5. Antibiotics

Antimicrobial agents remain indispensable for infection management in oncology patients, most notably those receiving chemotherapeutic or immunotherapeutic interventions, wherein immunocompromised states substantially elevate susceptibility to bacterial pathogens [123,124]. Nevertheless, the administration of broad-spectrum antimicrobials inevitably disrupts intestinal microbial ecosystems, potentially compromising the therapeutic efficacy of immune-mediated cancer treatments.

Observational clinical investigations suggest that recent antimicrobial exposure correlates with diminished responsiveness to ICI therapy and reduced survival duration, although considerable heterogeneity exists across research findings [125,126,127]. Experimental studies utilizing murine models demonstrated that antimicrobial administration depletes beneficial bacterial populations, particularly Bifidobacterium species, subsequently altering critical immunological signaling cascades that include T-lymphocyte activation, major histocompatibility complex class I expression, and SCFA production, ultimately attenuating ICI therapeutic efficacy [52,128].

Paradoxically, antimicrobial agents concurrently serve as strategic therapeutic modalities within oncological contexts when applied with precision. The eradication of *Helicobacter pylori* infection substantially mitigates gastric carcinoma risk [129], while selective antibiotic-based therapeutic approaches targeting tumor-associated bacterial communities have demonstrated the capacity to generate novel antigenic determinants and augment antitumor immunity [130].

Antimicrobial influence on ICI therapeutic outcomes has been extensively documented in pulmonary malignancies. Specific antibiotics—including ampicillin, colistin, and streptomycin—demonstrably attenuate PD-1 inhibitor efficacy, both as monotherapeutic agents and in combination with CTLA-4 blockade [5]. Comprehensive clinical analysis encompassing NSCLC, RCC, and urothelial carcinoma revealed that patients exposed to antimicrobial agents either preceding or during PD-1/PD-L1 targeted therapy experienced significantly abbreviated progression-free and overall survival intervals, suggesting that microbiome perturbation potentially contributes to ICI resistance mechanisms [5].

Achieving equilibrium between infectious disease control and the preservation of microbial diversity constitutes a fundamental challenge in contemporary cancer management. While extensive antimicrobial regimens potentially undermine ICI effectiveness, selective and judicious antimicrobial utilization—guided by optimal timing parameters and culture-directed selection protocols—may mitigate detrimental consequences while maintaining protective functions against infectious complications. Considering the intricate interactions between antimicrobial therapy, microbiome composition, and cancer treatment outcomes, individualized approaches to antimicrobial stewardship remain essential in optimizing patient care.

## 5. Conclusions

The microbiome has emerged as a critical component in the advancement of cancer therapeutics, particularly in immunotherapy and chemotherapy. Recent studies underscore the profound influence of the gut microbiota on treatment efficacy, immune modulation, and toxicity management. Specific microbial taxa and metabolites have been shown to enhance the effectiveness of ICIs and chemotherapeutic agents, while dysbiosis has been implicated in resistance mechanisms and adverse outcomes.

While many microbial taxa have been shown to promote therapeutic efficacy, others may compromise treatment responses or contribute to tumor persistence. This dual nature of the microbiome highlights the importance of considering both beneficial and harmful microbial components in the development of microbiome-based therapies.

Innovative approaches, such as engineered probiotics, prebiotics, and microbiome-modulating therapies, offer promising strategies to optimize treatment responses. Moreover, personalized microbiome profiling has opened new avenues for predicting therapeutic outcomes and tailoring interventions to individual patient needs. These findings highlight the potential of integrating microbiome-based strategies into precision oncology.

Despite significant progress, challenges remain, including the complexity of host–microbiome interactions, variability in microbiota composition, and the need for standardized methodologies. Future research should focus on elucidating the mechanisms underlying microbiome-mediated effects, developing robust biomarkers for patient stratification, and exploring the synergistic potential of combining microbiome-targeted therapies with existing cancer treatments.

By leveraging the microbiome’s role in cancer therapy, the medical community can unlock new opportunities to enhance therapeutic efficacy, minimize toxicity, and improve patient outcomes in the fight against cancer.

## Figures and Tables

**Figure 1 cancers-17-01252-f001:**
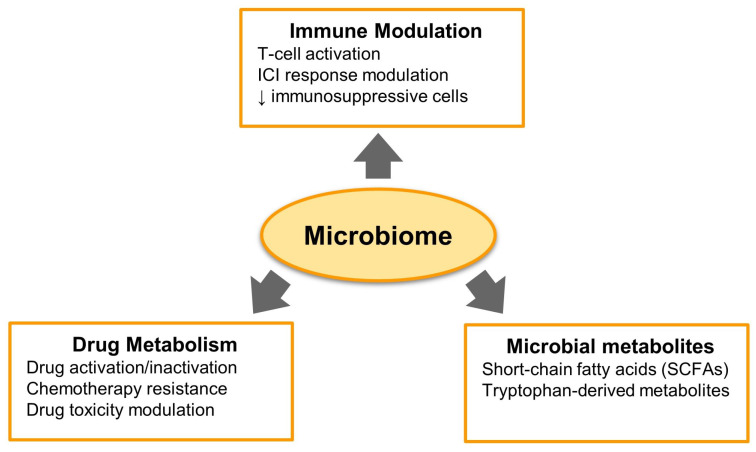
Mechanisms of microbiome influence in cancer therapy.

**Figure 2 cancers-17-01252-f002:**
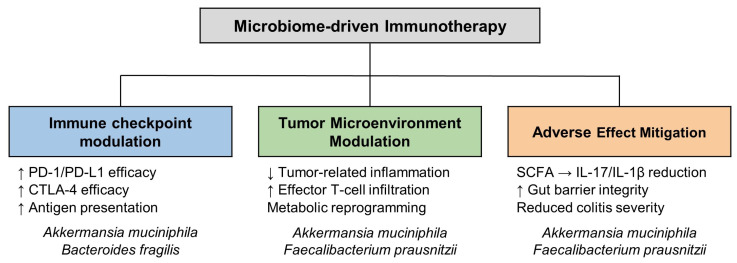
Microbiome-driven synergy with immunotherapy. Key microbial taxa, including *Akkermansia muciniphila*, *Faecalibacterium prausnitzii*, and *Bacteroides fragilis*, have demonstrated consistent associations with enhanced immune checkpoint inhibitor (ICI) efficacy. These effects are mediated through multiple pathways: (1) production of short-chain fatty acids (SCFAs), particularly butyrate, leading to histone deacetylase (HDAC) inhibition and increased T helper 1 (Th1) and cytotoxic T lymphocyte (CTL) responses; (2) inosine production that activates A2A receptors on T cells, promoting Th1 differentiation; (3) induction of interleukin-12 (IL-12) and enhancement of dendritic cell and CD8^+^ T cell activity; and (4) improvement of gut barrier integrity and modulation of systemic inflammation to mitigate immune-related toxicities.

**Figure 3 cancers-17-01252-f003:**
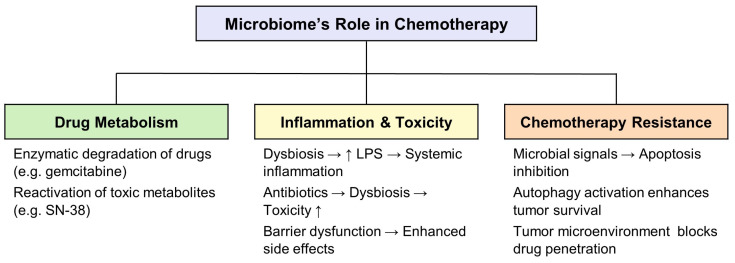
Microbiome-associated mechanisms contributing to chemotherapy response. Certain bacterial species, such as *Gammaproteobacteria*, degrade chemotherapeutic agents like gemcitabine via bacterial cytidine deaminase. Intestinal *Escherichia coli* can reactivate the toxic metabolite SN-38 from its inactive glucuronidated form (SN-38G), increasing gastrointestinal toxicity. Dysbiosis-induced barrier disruption permits translocation of lipopolysaccharide (LPS), exacerbating systemic inflammation and chemotoxicity. Additionally, bacteria such as *Fusobacterium nucleatum* promote chemoresistance by activating autophagy through the Toll-like receptor 4 (TLR4)/MyD88 pathway, thereby inhibiting apoptosis and limiting drug efficacy.

**Table 1 cancers-17-01252-t001:** Currently registered clinical trials on microbiome-modulated immunotherapy.

NCT No.	Cancer Type	Enrollment	Immunotherapy	Microbial Intervention	Phase	Location	Status
5865730	NSCLC, RCC	122	None (PD-1, PD-L1 agents)	Oral administration of Oncobax®-AK (Akkermansia muciniphila)	1/2	Belgium, France	Recruiting
6448572	NSCLC	21	Nivolumab	EXL01 (Faecalibacterium prausnitzii)	1/2	France	Recruiting
5487859	RCC	24	Ipilimumab, Nivolumab, Pembrolizumab, Lenvatinib, Everolimus, Cabozantinib	Acarbose	2	USA	Not yet recruiting
6551272	HCC	34	Atezolizumab, Bevacizumab	EXL01 (F. prausnitzii-based bacterial strain)	2	France	Not yet recruiting
2960282	Cancer	21	ICI	FMT	N/A	Switzerland	Completed
3891979	Pancreatic Adenocarcinoma	0	Pembrolizumab	Ciprofloxacin + Metronidazole	4	USA	Withdrawn
3595683	Melanoma	8	Pembrolizumab	Bifidobacterium longum EDP1503	2	USA	Active, not recruiting
3637803	NSCLC, RCC, Melanoma, Bladder Cancer	63	Pembrolizumab	MRx0518 (a lyophilized proprietary bacterium strain)	1/2	USA	Terminated
3686202	Solid tumors	65	Anti-PD-1/PD-L1	MET-4 (Microbial Ecosystem Therapeutics-4)	2/3	Canada	Active, not recruiting
3775850	CRC, Triple-negative breast cancer, NSCLC, Bladder, Gastroesophageal, RCC	69	Pembrolizumab	Bifidobacterium longum EDP1503	1	USA, Canada	Completed
3817125	Melanoma	14	Nivolumab	SER-401 (Live biotherapeutic products)	1	USA	Completed
3829111	RCC	30	Nivolumab + Ipilimumab	CBM588 (Butyricum CBM 588 probiotic strain)	1	USA	Completed
4208958	Melanoma, Gastric, Gastroesophageal Junction Adenocarcinoma, CRC	56	Nivolumab	VE800 (Live biotherapeutic products)	1/2	USA	Completed
4601402	Solid tumor, NSCLC, HNSCC, Urothelial Carcinoma	11	Avelumab	Live biotherapeutic product GEN-001	1	USA	Completed
4699721	NSCLC	60	Nivolumab + Paclitaxel + Carboplatin	BiFico (Bifidobacterium trifidum live powder)	1	China	Active, not recruiting
4909034	NSCLC	15	Pembrolizumab	MS-20 (Fermented Soybean Extract MicroSoy-20)	2	Taiwan	Completed
5032014	Liver	46	Anti-PD-1	Probio-M9 (Lactobacillus rhamnosus)	N/A	China	Unknown status
5094167	NSCLC	46	Carrilizumab + Platinum	Kex02 (Lactobacillus Bifidobacterium V9)	N/A	China	Unknown status
5122546	RCC	31	Nivolumab + Cabozantinib	CBM588 (C lostridium butyricum CBM 588 probiotic strain)	1	USA	Active, not recruiting
5220124	Bladder, Urothelial	190	Immunotherapy	Live combined Bifidobacterium, Lactobacillus and Enterococcus capsules	4	China	Unknown status
5354102	NSCLC, Melanoma, RCC	11	BMC128 (live bio-therapeutic product composed of 4 commensal bacterial strains)	BMC128	1	Israel	Active, not recruiting
5620004	Advanced HCC	30	Carrilizumab + Apatinib Mesylate	Bifidobacterium bifidum	1/2	China	Unknown status
5083416	HNSCC	29	Nivolumab, Pembrolizumab, Atezolizumab, Avelumab, or Durvalumab	Prolonged nightly fasting	N/A	USA	Completed

*Note:* This table summarizes registered clinical trials investigating microbiome-modulating strategies in conjunction with cancer immunotherapies. Trials are listed with cancer types, enrollment numbers, immune checkpoint inhibitors or other immunotherapy agents used, the type of microbial intervention applied, phase, study location, and trial status (as of 18 March 2025). Trial identifiers (NCT numbers) are hyperlinked to their respective ClinicalTrials.gov pages. *Abbreviations:* NSCLC—non-small-cell lung cancer; RCC—renal cell carcinoma; HCC—hepatocellular carcinoma; CRC—colorectal cancer; HNSCC—head and neck squamous cell carcinoma; ICI—immune checkpoint inhibitor; FMT—fecal microbiota transplantation; PD-1—programmed death-1; PD-L1—programmed death-ligand 1.

**Table 2 cancers-17-01252-t002:** Currently registered clinical trials on microbiome-modulated chemotherapy.

NCT No.	Cancer Type	Enrollment	Chemotherapy	Microbial Intervention	Phase	Location	Status
2928523	Acute myeloid leukaemia	20	Induction chemotherapy	Single-arm: autologous FMT from pre-chemotherapy	1/2	France	Completed
2771470	Lung cancer	41	Initiating chemotherapy	RCT: Clostridium butyricum probiotic vs. placebo	1	China	Completed
3314688	Breast cancer	173	Initiating chemotherapy	RCT: ACS recommended diet + exercise guidelines	N/A	USA	Active, not recruiting
1410955	CRC	46	Initiating irinotecan	RCT: colon Dophilus probiotic vs. placebo	3	Slovakia	Completed
2944617	Metastatic kidney cancer	21	Initiating TKIs	RCT: Activia yogurt (Bifidobacterium lactis)	N/A	USA	Completed
2819960	CRC	233	Initiating irinotecan	RCT: PROBIO-FIX INUM probiotic vs. placebo	3	Slovakia	Completed
197873	CRC	84	Initiating capecitabine, oxaliplatin	RCT: Lactobacilli (GefilusR) vs. placebo	N/A	Finland	Completed
3642548	NSCLC	180	Initiating platinum-based chemotherapy	RCT: Bifico vs. placebo	3	China	Unknown
3705442	CRC	76	Treated with FOLFIRI	RCT: Omni-Biotic 10 vs. placebo	2	Croatia	Unknown
4021589	CRC	40	Chemotherapy	Weileshu (probiotics)	2	China	Completed
3870607	Anal canal squamous cell cancer	75	Chemoradiotherapy (Ch-RT)	Prebiotics + probiotics	2	Brazil	Unknown

*Note:* This table presents registered clinical trials investigating microbial interventions in conjunction with chemotherapy for various cancer types retrieved from ClinicalTrials.gov as of 18 March 2025. Trials are categorized by cancer type, enrollment size, chemotherapeutic regimens (including irinotecan-, platinum-, or capecitabine-based protocols), type of microbial intervention (e.g., probiotics, prebiotics, dietary modifications, or fecal microbiota transplantation), study phase, location, and trial status (as of 18 March 2025). Trial identifiers (NCT numbers) are hyperlinked to their respective ClinicalTrials.gov pages. *Abbreviations:* CRC—colorectal cancer; NSCLC—non-small-cell lung cancer; FMT—fecal microbiota transplantation; RCT—randomized controlled trial; TKIs—tyrosine kinase inhibitors; FOLFIRI—folinic acid, fluorouracil, and irinotecan; ACS—American Cancer Society; PROBIO-FIX INUM—probiotic supplement containing Bifidobacterium animalis subsp. lactis BB-12® and Lactobacillus rhamnosus GG® (LGG®)

**Table 3 cancers-17-01252-t003:** Currently registered clinical trials of FMT modulating the gut microbiome in immunotherapy.

NCT No.	Cancer Type	Enrollment	Treatment	Microbial Intervention	Phase	Location	Status
5502913	Lung Cancer	80	Immune Checkpoint Inhibitors	FMT	2	Israel	Recruiting
5251389	Melanoma	24	Anti-PD-1	FMT	1/2	Netherlands	Recruiting
3819296	Melanoma, Genitourinary, Malignant Solid Neoplasm	800	ICIs	FMT from healthy donors	2	USA	Recruiting
4038619	Genitourinary, Melanoma, Lung, Ovary, Uterus, Breast, Cervical	40	Loperamide	FMT via colonoscopy	1	USA	Recruiting
6205862	Colorectal Adenoma	466	None	FMT	2	China	Recruiting
4975217	PDAC	10	None	FMT	1	USA	Recruiting
4988841	Melanoma	60	Ipilimumab + Nivolumab	Fecal microbiotherapy (MaaT013)	2	France	Recruiting
3772899	Melanoma	20	Pembrolizumab/Nivolumab	FMT capsules from healthy donors	1	Canada	Active, not recruiting
4163289	RCC	20	Ipilimumab + Nivolumab	FMT capsules	1	Canada	Active, not recruiting
4729322	Metastatic CRC	15	ICIs	FMT capsules	1/2	Italy	Active, not recruiting
4951583	NSCLC, Melanoma	45	Pembrolizumab, Ipilimumab + Nivolumab	Investigational FMT	2	Canada	Active, not recruiting
5273255	Malignancies	18	ICIs	FMT via endoscopy	N/A	Switzerland	Completed
4924374	Lung Cancer	25	Pembrolizumab, Atezolizumab	FMT capsules	N/A	Spain	Completed
3341143	Melanoma	20	Pembrolizumab	FMT via colonoscopy from ICI responders	2	USA	Completed
4056026	Mesothelioma	1	Pembrolizumab (Keytruda)	Single-dose FMT infusion	1	USA	Completed
4130763	GI System Cancer	10	Anti-PD-1	FMT capsules	1	China	Completed
3353402	Melanoma	40	Anti-PD-1	FMT capsules from ICI responders	1	Israel	Unknown
4116775	Prostate Cancer	32	Pembrolizumab	FMT via endoscopy	2	USA	Unknown
4264975	Solid Carcinoma	60	Immunotherapy	FMT	N/A	Korea	Unknown
4521075	Melanoma, NSCLC	42	Nivolumab	FMT capsules	1/2	Israel	Unknown
5008861	NSCLC	20	Anti-PD-1/PD-L1	FMT capsules	1	China	Unknown
4577729	Malignant Melanoma	5	Pembrolizumab/Nivolumab	Allogenic FMT; Autologous FMT	N/A	Austria	Terminated
3812705	Hematopoietic and Lymphoid Cell Neoplasm	6	N/A	FMT	2	China	Completed
5669846	NSCLC	26	Pembrolizumab	Responder-derived FMT (R-FMT)	2	USA	Recruiting
5690048	HCC	48	Atezolizumab + Bevacizumab	FMT	2	Germany	Not yet recruiting

*Note:* This table summarizes ongoing and completed clinical trials investigating the use of fecal microbiota transplantation (FMT) to modulate the gut microbiome in the context of cancer treatment. Included trials vary in cancer types, immunotherapy regimens (e.g., immune checkpoint inhibitors), and methods of FMT delivery (e.g., capsules, colonoscopy, or endoscopy). Trials also differ in donor source, such as healthy individuals or prior responders to immunotherapy. Key trial information includes cancer type, enrollment size, treatment, microbial intervention strategy, study phase, geographic location, and trial status (as of 18 March 2025). Trial identifiers (NCT numbers) are hyperlinked to their respective ClinicalTrials.gov pages. *Abbreviations:* FMT—fecal microbiota transplantation; ICI—immune checkpoint inhibitor; NSCLC—non-small-cell lung cancer; PDAC—pancreatic ductal adenocarcinoma; RCC—renal cell carcinoma; CRC—colorectal cancer; GI—gastrointestinal; HCC—hepatocellular carcinoma; PD-1—programmed death-1; PD-L1—programmed death-ligand 1.

## Data Availability

The data that are discussed in this article are presented in cited studies.

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
