# Peer review of "Optimizing Cancer Treatment Through Gut Microbiome Modulation"

_cancers, 2025, doi:10.3390/cancers17071252_

Round 1

Reviewer 1 Report

Comments and Suggestions for Authors

The gut microbiome plays a crucial role in the modulation of cancer therapies, including immunotherapy and chemotherapy. Recent evidence indicates that the gut microbiome impacts therapeutic efficacy, immune responses, and resistance mechanisms. Specific microbial taxa have been shown to enhance the efficacy of immune checkpoint inhibitors, while dysbiosis may contribute to adverse outcomes. The effects of chemotherapy are similarly influenced by microbial composition, with engineered probiotics and prebiotics presenting promising strategies for enhancing drug delivery and reducing toxicity. Microbial metabolites, such as short-chain fatty acids, and engineered microbial systems show potential for improving therapeutic response. These findings underscore the significance of personalized microbiome approaches in optimizing cancer therapies. This review is well crafted and can be accepted following minor revisions, which are outlined below.

  1. The discourse regarding potential safety concerns and adverse effects associated with microbiome interventions, including infection risk and allergic reactions, warrants amplification. Enhancing this discussion will facilitate a more comprehensive understanding of the clinical application prospects for these interventions.
  2. The impact of individual microbiome variations on treatment outcomes warrants discussion. To comprehend its dynamic characteristics, fluctuations in the microbiome at various time points were examined.
  3. The discussion should address the reversibility of microbiome alterations and their enduring impact on cancer treatment, in order to facilitate the development of sustainable treatment strategies and mitigate potential long-term side effects.
  4. Authors should comprehensively examine and discuss the impacts of carcinogenic and complicit microbes, as detailed in the following literature. (M.-Y. Li, A. Gu, J. Li, N. Tang, M. Matin, Y. Yang, G. Zengin, A. G. Atanasov. Exploring food and medicine homology: potential implications for cancer treatment innovations. Acta Materia Medica 2025, 4, 200-206. DOI: 10.15212/AMM-2025-0003)
  5. It is recommended to cite the following literature

[1] Liu Y-M, Liu C, Deng Y-S, et al. Beneficial effects of dietary herbs on high-fat diet-induced obesity linking with modulation of gut microbiota. Food & Medicine Homology, 2025, 2(2): 9420034. https://doi.org/10.26599/FMH.2025.9420034

[2] Zhang Z, Gao Q, Ren X, et al. Characterization of intratumor microbiome in cancer immunotherapy. The Innovation, 2023, 4(5), 100482. https://doi.org/10.1016/j.xinn.2023.100482

[3] L. Yang, N. Li, X. Yi, Z. Wang, The translational potential of the lung microbiome as a biomarker and a therapeutic target for chronic obstructive pulmonary disease. Interdiscip. Med. 2023, 1, e20230023. https://doi.org/10.1002/INMD.20230023

Author Response

Comment 1: The discourse regarding potential safety concerns and adverse effects associated with microbiome interventions, including infection risk and allergic reactions, warrants amplification. Enhancing this discussion will facilitate a more comprehensive understanding of the clinical application prospects for these interventions. 

Response 1: We appreciate the reviewer’s insightful comment regarding safety concerns and adverse effects of microbiome interventions. In response, we have expanded the discussion on 4.1. FMT and 4.3. Probiotics to include infection risks, donor variability, and biosafety issues such as ARG transfer and regulatory challenges, supported by recent literature. 

Comment 2: The impact of individual microbiome variations on treatment outcomes warrants discussion. To comprehend its dynamic characteristics, fluctuations in the microbiome at various time points were examined. 

Response 2: We appreciate the reviewer’s comment regarding the impact of individual microbiome variation and the importance of capturing its dynamic characteristics. In response, we have revised the 2.5. metastatic melanoma and 3.2. lung cancer sections to include evidence from longitudinal studies highlighting temporal microbiome fluctuations during therapy, as well as interindividual variability affecting treatment outcomes and biomarker reliability. 

Comment 3: The discussion should address the reversibility of microbiome alterations and their enduring impact on cancer treatment, in order to facilitate the development of sustainable treatment strategies and mitigate potential long-term side effects. 

Response 3: Thank you for the insightful suggestion regarding the reversibility of microbiome alterations and their enduring impact on cancer treatment. To address this point, we have incorporated new content across multiple sections to reflect the longitudinal dynamics and potential stability of microbiome shifts in the context of therapy. In the 2.5. metastatic melanoma section, we added recent evidence showing that certain beneficial taxa remain elevated even after ICI therapy cessation in responders, whereas transient expansions in non-responders tend to regress post-treatment (Björk et al., 2024). In the 3.1. GI cancer section, we incorporated findings demonstrating the re-emergence of beneficial genera such as Bifidobacterium and Faecalibacterium following chemotherapy in responders, supporting the notion of partial ecological recovery (Li et al., 2020). 
Finally, in the 4.1. FMT section, we added a statement highlighting the sustained presence of donor-derived strains up to three months post-transplantation, as confirmed by strain-level tracking (Xu et al., 2022). 

Comment 4: Authors should comprehensively examine and discuss the impacts of carcinogenic and complicit microbes, as detailed in the following literature. (M.-Y. Li, A. Gu, J. Li, N. Tang, M. Matin, Y. Yang, G. Zengin, A. G. Atanasov. Exploring food and medicine homology: potential implications for cancer treatment innovations. Acta Materia Medica 2025, 4, 200-206. DOI: 10.15212/AMM-2025-0003) 

Respone 4: We thank the reviewer for pointing out the importance of addressing the dual role of the microbiome in cancer. In response, we have expanded our discussion to incorporate the concept of complicit microbes that promote tumorigenesis through mechanisms such as chronic inflammation, immune modulation, and pro-oncogenic signaling. Specifically, we revised the 3.1. gastrointestinal cancer section.  

Comment 5: It is recommended to cite the following literature 

[1] Liu Y-M, Liu C, Deng Y-S, et al. Beneficial effects of dietary herbs on high-fat diet-induced obesity linking with modulation of gut microbiota. Food & Medicine Homology, 2025, 2(2): 9420034. https://doi.org/10.26599/FMH.2025.9420034 

[2] Zhang Z, Gao Q, Ren X, et al. Characterization of intratumor microbiome in cancer immunotherapy. The Innovation, 2023, 4(5), 100482. https://doi.org/10.1016/j.xinn.2023.100482 

[3] L. Yang, N. Li, X. Yi, Z. Wang, The translational potential of the lung microbiome as a biomarker and a therapeutic target for chronic obstructive pulmonary disease. Interdiscip. Med. 2023, 1, e20230023. https://doi.org/10.1002/INMD.20230023 

Respone 5: We thank the reviewer for the thoughtful suggestions. In response, we have made several revisions to enhance the manuscript’s clarity and scientific depth. 

In the 4.2. prebiotics section, we introduced recent insights into the modulatory effects of plant-derived compounds on gut microbiota, particularly their ability to enrich beneficial, SCFA-producing taxa associated with anti-inflammatory and metabolic regulatory functions. 

In the 2.5. metastatic melanoma section, we expanded our discussion of the intratumor microbiome by highlighting its association with immune infiltration and immunotherapy responsiveness. Recent studies suggest that certain intratumoral taxa may enhance CD8⁺ T cell recruitment and cytotoxic activity, thereby shaping the tumor-immune microenvironment in a clinically meaningful way. 

Additionally, in the 2.2. lung cancer section, we contextualized the role of the lung microbiome by drawing on findings from chronic respiratory conditions. 

We hope these revisions improve the comprehensiveness and translational relevance of our discussion. 

Reviewer 2 Report

Comments and Suggestions for Authors

The review of Kim et al. entitled “Optimizing Cancer Treatment Through Microbiome Modulation” is a very good summary of current knowledge and in progress studies concerning the impact of the microbiome on immunotherapy (part 2) and chemotherapy strategies (part 3) for different types of cancer (gastrointestinal, lung, liver, melanoma) throughout the world. The authors also examine the impact of fecal microbiota transplantation as well as prebiotics, probiotics, post-biotics and antibiotics consumption on cancer treatment (part 4).
The described approaches and their outcomes are still limited but seem promising. The text is clear and pleasant to read.
However some aspects of the manuscript could be improved.

  • References to support authors assertions should be provided in the introduction section as well as in other places in the manuscript (lines, 49, 53, 75, 79 etc…)
  • A list of the numerous abbreviations used should be provided at the end of the manuscript.
  • In all the Tables listing the various clinical trials in various countries, it would be nice to have the web site of the Institutions that are doing these assays.
  • It would also be nice to have a specific section describing the microbiome characteristics that are correlating with specific cancers. Is dysbiosis the cause or the consequence of cancer?
    For instance, what is the impact of tobacco on lung microbiota and what is its relation with lung cancer.
  • It would also be nice to have a specific section describing the known consequences of immunotherapy and chemotherapy strategies on the microbiome. The authors report the correlation between the presence of specific phylum and the characteristics of responders / non-responders to a treatment. Is it possible to know whether specific phylum were present and abundant before the treatment or were selected / eliminated by the treatment.
  • When a specific bacteria is administrated it would be nice to mention whether that is an oral or fecal administration…
  • What are the “neoadjuvant chemotherapeutic approaches”? (P 10)
  • - Line 415: 70% (?).

Author Response

Comment 1: References to support authors assertions should be provided in the introduction section as well as in other places in the manuscript (lines, 49, 53, 75, 79 etc…) 

Response 1: We thank the reviewer for pointing out the need for appropriate referencing throughout the manuscript. In response, we have added citations to support the statements on lines 49 and 53, and have carefully reviewed the entire manuscript to ensure that all key assertions are adequately referenced. Additionally, lines 75 and 79 were revised during the editing process, and the updated content is now supported by relevant citations. We appreciate the reviewer’s attention to detail, which has helped improve the rigor and clarity of the manuscript. 

Comment 2: A list of the numerous abbreviations used should be provided at the end of the manuscript. 

Response 2: We appreciate the reviewer’s suggestion. In response, we have added a dedicated "Abbreviations" section at the end of the manuscript, presenting a table that lists all abbreviations along with their full definitions. 

Comment 3: In all the Tables listing the various clinical trials in various countries, it would be nice to have the web site of the Institutions that are doing these assays. 

Response 3: Thank you for your suggestion. We have revised Tables 1, 2, and 3 to include direct hyperlinks for each clinical trial number, leading to the corresponding ClinicalTrials.gov page. This allows readers to easily access up-to-date and detailed information about the institutions conducting these trials, as well as other relevant trial details. 

Comment 4: It would also be nice to have a specific section describing the microbiome characteristics that are correlating with specific cancers. Is dysbiosis the cause or the consequence of cancer? 
For instance, what is the impact of tobacco on lung microbiota and what is its relation with lung cancer. 

Response 4: We appreciate the reviewer’s suggestion to address cancer-specific microbiome characteristics and the causal versus consequential nature of dysbiosis. In response, we have expanded the introduction of Section 4 to discuss current perspectives on the bidirectional relationship between dysbiosis and cancer. Additionally, in the 2.2. lung cancer section, we incorporated recent findings on tobacco-induced microbial alterations and their potential immunological implications.  

Comment 5: It would also be nice to have a specific section describing the known consequences of immunotherapy and chemotherapy strategies on the microbiome. The authors report the correlation between the presence of specific phylum and the characteristics of responders / non-responders to a treatment. Is it possible to know whether specific phylum were present and abundant before the treatment or were selected / eliminated by the treatment. 

Response 5: We thank the reviewer for this insightful comment. In response, we have incorporated longitudinal evidence into the 2.5. metastatic melanoma and 3.2. lung cancer sections to highlight how microbiome composition can change during immunotherapy and chemotherapy. These additions aim to address the question of whether specific taxa are pre-existing or selected by treatment. However, we acknowledge that current literature remains limited in definitively establishing causality or temporal directionality, and further studies are needed to clarify these dynamics.  

Comment 6: When a specific bacteria is administrated it would be nice to mention whether that is an oral or fecal administration… 

Response 6: We appreciate the reviewer’s suggestion to clarify the route of bacterial administration. Accordingly, we have revised the relevant sentences to indicate that the bacterial interventions discussed were delivered orally. 

Comment 7: What are the “neoadjuvant chemotherapeutic approaches”? (P 10) 

Response 7: We thank the reviewer for pointing out the need to clarify the term “neoadjuvant chemotherapeutic approaches.” In response, we have added a brief explanation to define neoadjuvant chemotherapy.  

Comment 8: - Line 415: 70% (?). 

Response 8: We appreciate your careful review. The sentence was unintentionally truncated in an earlier version, but it has since been removed during content revision and no longer appears in the current manuscript. 

Reviewer 3 Report

Comments and Suggestions for Authors

Dear Authors,

The manuscript is well written and structured. However some details need to be improved before publication:

  • What specific microbial taxa have emerged as the most influential in modulating immune checkpoint inhibitor efficacy across different cancer types, and through which molecular pathways do they act?
  • How can the dual roles of microbial metabolites—such as short-chain fatty acids—in both enhancing immune responses and contributing to treatment resistance be disentangled to inform personalized therapy?
  • In what ways do engineered probiotics and prebiotics compare in their ability to enhance chemotherapy efficacy and mitigate toxicity, and what are the primary barriers to clinical translation of these interventions?
  • Given the variability in donor microbiota profiles, what strategies (including synthetic microbial consortia) show the most promise in standardizing fecal microbiota transplantation (FMT) protocols for cancer patients?
  • How does the interplay between gut and lung microbiota—the gut–lung axis—differ in its impact on tumor microenvironment modulation in lung cancer versus gastrointestinal cancers?
  • Which microbiome-derived biomarkers have the greatest potential for predicting patient responsiveness to immunotherapies, and how might these be integrated into existing clinical decision-making processes?
  • What are the underlying mechanisms by which antibiotic administration prior to or during cancer treatment modulates treatment outcomes, and what measures can be taken to mitigate negative impacts on therapeutic efficacy?
  • How can advanced computational models and metagenomic profiling be refined to better capture the complexity of host–microbiome interactions and inform precision oncology strategies?
  • In the context of liver cancer, how do alterations in bile acid metabolism driven by dysbiosis contribute to chemotherapy resistance, and what targeted interventions could restore metabolic balance?
  • What novel insights does the study provide regarding the use of postbiotics for modulating the tumor microenvironment, and how might these findings guide the development of next-generation adjunctive cancer therapies?

Author Response

Comment 1: What specific microbial taxa have emerged as the most influential in modulating immune checkpoint inhibitor efficacy across different cancer types, and through which molecular pathways do they act? 

Respone 1:  Thank you for this important comment. To clarify the key microbial taxa and their mechanistic roles in modulating therapeutic responses, we have expanded the captions of Figure 2 and Figure 3 to include specific bacterial species and representative molecular pathways involved in immunotherapy and chemotherapy contexts, respectively. We believe this provides a clearer and more concise summary of the microbiome-derived mechanisms discussed in the manuscript. 

Comment 2: How can the dual roles of microbial metabolites—such as short-chain fatty acids—in both enhancing immune responses and contributing to treatment resistance be disentangled to inform personalized therapy? 

Response 2: Thank you for this valuable suggestion. To address your comment, we revised Section 2.1 to include recent evidence showing that SCFAs may promote tumor progression under specific microbial and host contexts. We also added supporting epidemiological data linking certain circulating SCFA levels with increased CRC risk . In the 4.4. Postbiotics section, we further incorporated findings that highlight the immunosuppressive effects of SCFAs, including Treg expansion and IL-10 production, which depend on host genotype, metabolite concentration, and immune context .  

Comment 3: In what ways do engineered probiotics and prebiotics compare in their ability to enhance chemotherapy efficacy and mitigate toxicity, and what are the primary barriers to clinical translation of these interventions? 

Respone 3: Thank you for your insightful feedback. In response, we have added a comparison between engineered probiotics and prebiotics in the 4.3. Probiotics section, emphasizing that prebiotics depend on host-driven microbial fermentation and show interindividual variability, whereas engineered probiotics enable targeted delivery of therapeutic molecules like IL-2 within the tumor microenvironment, offering greater precision. 

We have also expanded the discussion of clinical translation barriers, including safety concerns (e.g., bacteremia, ARG transfer), variability in strain viability and colonization, and regulatory challenges limiting the application of probiotic-based therapies. 

Comment 4: Given the variability in donor microbiota profiles, what strategies (including synthetic microbial consortia) show the most promise in standardizing fecal microbiota transplantation (FMT) protocols for cancer patients? 

Response 4: We appreciate the reviewer’s valuable suggestion regarding the need to address donor microbiota variability and standardization strategies in the context of fecal microbiota transplantation (FMT). In response, we have revised the 4.1. FMT section to clarify that donor-specific microbial composition critically influences FMT outcomes and to highlight recent approaches—such as synthetic microbial consortia and microbial fingerprinting—that aim to overcome this limitation.  

Comment 5: How does the interplay between gut and lung microbiota—the gut–lung axis—differ in its impact on tumor microenvironment modulation in lung cancer versus gastrointestinal cancers? 

Response 5: We appreciate the reviewer’s insightful comment. In response, we have clarified the distinct mechanisms through which the microbiota modulates the tumor microenvironment in lung versus gastrointestinal cancers. Specifically, in the revised subsection on lung cancer immunotherapy (Section 2.2), we now highlight that while microbial regulation in gastrointestinal tumors occurs primarily via local mucosal immune signaling, lung tumors are predominantly influenced by systemic immune pathways mediated through the gut–lung axis.  

Comment 6: Which microbiome-derived biomarkers have the greatest potential for predicting patient responsiveness to immunotherapies, and how might these be integrated into existing clinical decision-making processes? 

Response 6: We thank the reviewer for this important comment. In response, we have revised Section 2.5 (Metastatic melanoma) to highlight Akkermansia muciniphila and Faecalibacterium prausnitzii as the most promising microbiome-derived biomarkers of ICI responsiveness. These taxa are emphasized based on their consistent enrichment in responders, mechanistic validation, and ongoing clinical evaluation as adjunctive therapies (e.g., Oncobax-AK and EXL01). We also address their integration into predictive models and their potential utility in guiding ICI selection and personalization. 

Comment 7:  What are the underlying mechanisms by which antibiotic administration prior to or during cancer treatment modulates treatment outcomes, and what measures can be taken to mitigate negative impacts on therapeutic efficacy? 

Respone 7: We appreciate the reviewer’s insightful comment regarding the mechanistic effects of antibiotic exposure and strategies to mitigate its negative impact on immunotherapy efficacy. While the original manuscript addressed key immunological mechanisms (e.g., loss of Bifidobacterium, reduced T-cell activation and SCFA production), we have revised the relevant section to explicitly incorporate microbiome-restorative strategies—such as prebiotics, probiotics, dietary modulation, and fecal microbiota transplantation—supported by recent population-level evidence (Eng et al., JCO 2023). These changes are intended to more fully address the reviewer’s concern while preserving conciseness. 

Comment 8: How can advanced computational models and metagenomic profiling be refined to better capture the complexity of host–microbiome interactions and inform precision oncology strategies? 

Response 8: We thank the reviewer for this insightful comment. In response, we have clarified the importance of strain-level engraftment predictability and donor–recipient microbial compatibility in shaping FMT outcomes, citing recent findings that support the development of predictive frameworks based on metagenomic data (Schmidt et al., 2022). In addition, we have discussed microbial fingerprinting as an emerging strategy to optimize donor selection (Servetas et al., 2022), aligning with the reviewer's suggestion to highlight refined profiling approaches that support precision interventions. These revisions are integrated in the updated 4.1. FMT subsection. 

Comment 9: In the context of liver cancer, how do alterations in bile acid metabolism driven by dysbiosis contribute to chemotherapy resistance, and what targeted interventions could restore metabolic balance? 

Response 9: We thank the reviewer for this insightful comment. In Section 3.3, we have revised the discussion on liver cancer chemotherapy to incorporate a mechanistic explanation of how dysbiosis-induced disruption of bile acid metabolism, particularly the depletion of conjugated secondary bile acids such as glycodeoxycholic acid (GDCA), contributes to chemotherapy resistance. We have also added relevant preclinical evidence supporting targeted interventions. 

Comment 10: What novel insights does the study provide regarding the use of postbiotics for modulating the tumor microenvironment, and how might these findings guide the development of next-generation adjunctive cancer therapies? 

Respone 10: Thank you for your valuable comment. In response, we have clarified the novel mechanistic insights of postbiotics—particularly their colonization-independent effects—and discussed how these features may inform the development of targeted adjunctive strategies in cancer therapy. Relevant findings from recent studies have been incorporated to support this point (see revised subsection 4.4. Postbiotics). 

Reviewer 4 Report

Comments and Suggestions for Authors

Thank you for the opportunity to review your narrative review on the Microbiome and Cancer treatment. Please see my feedback below:

Overall, the paper was nicely organized into clear sections, and accessible for the reader.

Section 2.1: Is lacking details on specific studies with data that have been provided in some other sections. All of the sections should include data from the work they are citing.

Tables. More comprehensive figure legends should be provided, clarifying abbreviations etc.

Ln. 415, abrupt end to sentence.

Explicitly clarify what is meant by pre-, pro- and post- in terms of timing and aims.

There is no mention of the Oral Microbiome. This should at least be mentioned, as several oral microbes have been associated with cancer. Additionally, the title of the paper should be changed to reflect that it only covers the gut microbiome.

Author Response

Comment 1: Section 2.1: Is lacking details on specific studies with data that have been provided in some other sections. All of the sections should include data from the work they are citing. 

Respone 1: We appreciate the reviewer’s insightful comment regarding the need for more data-driven and structured content in Section 2.1. In response, we have substantially revised the section to ensure that each claim is supported by quantitative findings from the cited studies. Specifically, we incorporated relevant concentrations, statistical values (e.g., OR, q-values, Cohen’s d), and experimental conditions (e.g., murine models, T cell subsets) to clarify the evidence base. We also refined the paragraph transitions and restructured the logical flow to reflect the progression from microbial metabolites and structural components to their immunologic implications across CRC and other GI cancers.  

Comment 2: Tables. More comprehensive figure legends should be provided, clarifying abbreviations etc. 

Response 2: We appreciate the reviewer’s suggestion regarding the table legends. We have revised the legends for Tables 1–3 to provide more comprehensive descriptions, including clarification of abbreviations, study characteristics, and methodological details where appropriate. 

Comment 3: Ln. 415, abrupt end to sentence. 

Response 3: We appreciate your careful review. The sentence was unintentionally truncated in an earlier version, but it has since been removed during content revision and no longer appears in the current manuscript. 

Comment 4: Explicitly clarify what is meant by pre-, pro- and post- in terms of timing and aims. 

Response 4: We thank the reviewer for this insightful suggestion. In response, we have revised the introduction of the “4. Microbial Intervention” section to explicitly differentiate prebiotics, probiotics, and postbiotics in terms of their timing of administration and therapeutic objectives.  

Comment 5: There is no mention of the Oral Microbiome. This should at least be mentioned, as several oral microbes have been associated with cancer.  

Response 5: We appreciate the reviewer’s suggestion to address the role of the oral microbiome. In response, we have incorporated relevant content into the Lung Cancer – Immunotherapy subsection, highlighting the contribution of oral commensals such as Streptococcus and Veillonella to immune modulation in the lower airways.  

Comment 6: Additionally, the title of the paper should be changed to reflect that it only covers the gut microbiome 

Respone 6: We appreciate the reviewer’s suggestion regarding the scope of the manuscript title. To more accurately reflect the content, we have revised the title to “Optimizing Cancer Treatment Through Gut Microbiome Modulation.” 

Round 2

Reviewer 3 Report

Comments and Suggestions for Authors

Dear authors,

I recommend the publication of the manuscript in the present form, considering that all the reviewers' comments were answered.